# Acute Mountain Sickness and the Risk of Subsequent Psychiatric Disorders—A Nationwide Cohort Study in Taiwan

**DOI:** 10.3390/ijerph20042868

**Published:** 2023-02-06

**Authors:** Ya-Hsuan Wang, Wu-Chien Chien, Chi-Hsiang Chung, Yu-Ning Her, Chia-Yi Yao, Biing-Luen Lee, Fang-Ling Li, Fang-Jung Wan, Nian-Sheng Tzeng

**Affiliations:** 1Department of Psychiatry, Tri-Service General Hospital, School of Medicine, National Defense Medical Center, Taipei City 11490, Taiwan; 2Department of Medical Research, Tri-Service General Hospital, National Defense Medical Center, Taipei City 11490, Taiwan; 3School of Public Health, National Defense Medical Center, Taipei City 11490, Taiwan; 4Taiwanese Injury Prevention and Safety Promotion Association, Taipei City 11490, Taiwan; 5Graduate Institute of Life Sciences, National Defense Medical Center, Taipei City 11490, Taiwan; 6Department of Plastic Surgery, Yonghe Cardinal Tien Hospital, New Taipei City 23148, Taiwan; 7Department of Medical Research, Tri-Service General Hospital, Beitou Branch, National Defense Medical Center, Taipei City 11243, Taiwan; 8Student Counseling Center, National Defense Medical Center, Taipei City 11490, Taiwan

**Keywords:** acute mountain illness, psychiatric disorders, National Health Insurance Research Database

## Abstract

We aim to explore if there is a relationship between acute mountain sickness (AMS) and the risk of psychiatric disorders in Taiwan by using the National Health Insurance Research Database for to the rare studies on this topic. We enrolled 127 patients with AMS, and 1270 controls matched for sex, age, monthly insured premiums, comorbidities, seasons for medical help, residences, urbanization level, levels of care, and index dates were chosen from 1 January 2000 to 31 December 2015. There were 49 patients with AMS and 140 controls developed psychiatric disorders within the 16-year follow-up. The Fine–Gray model analyzed that the patients with AMS were prone to have a greater risk for the development of psychiatric disorders with an adjusted sub-distribution hazard ratio (sHRs) of 10.384 (95% confidence interval [CI]: 7.267–14.838, *p* < 0.001) for psychiatric disorders. The AMS group was associated with anxiety disorders, depressive disorders, bipolar disorder, sleep disorders, posttraumatic stress disorder/acute stress disorder, psychotic disorder, and substance-related disorder (SRD). The relationship between anxiety, depression, sleep disorders, SRD, and AMS still persisted even after we excluded the psychiatric disorders within the first five years after AMS. There was an association between AMS and the rising risk of psychiatric disorders in the 16 years of long-term follow-up research.

## 1. Introduction

It is now much more popular to partake in mountain hiking at high altitudes [1]. Some of the individuals, who only stay in the lower altitude, fall victim to acute mountain sickness (AMS) or high altitude illness (HAI) after fast trekking to high altitudes [2]. The essential cause of AMS is low-barometric hypoxia at high altitudes. It is mentally and physically challenging and risky when being exposed to high altitudes [3]. Additionally, human beings had a long history in activities in the high-altitude areas, such as forestry and grazing [4]. Additionally, Hüfner et al., (2022) suggested that individuals with pre-existing psychiatric conditions were prone to have an elevated risk of AMS [5]. Furthermore, AMS is a hidden problem for army forces and their exercises [6], as many of the conflict war zones are in mountainous areas [7]. Therefore, the high-altitude exposure is also important for both civilian and military lives.

In Taiwan, one study found that, in three travel spots (Ho-Hwan Mountain: 3050 m meter, Yu Mountain: 2600 m, and Tai-Ping Mountain-1920 m), 381 trekkers with 160 (42%) met the diagnosis of AMS, between November 2000 and June 2001 [8]. In addition, there were four studies about the AMS on the highest mountain in Taiwan, the Jade Mountain (altitude: 3952 m): one study, between April 2007 and March 2008, found that 36% (N = 384) in 1066 trekkers on the Paiyun Lodge suffered from AMS, and the incidence of AMS on Jade Mountain varied by month, and associated with climbing experience, pre-exposure, and past history of AMS [9]. Other research for 89 trekkers on the Jade Mountain revealed that 27% of the trekkers met the criteria for AMS [10]. Furthermore, one prospective study, between 18 October and 27 October 2008, revealed a total of 787 subjects eligible for analysis, 32.2% of whom met the diagnosis of AMS [11]. In a prospective cohort study that included a total of 96 healthy non-acclimatized children aged 11–12 years who trekked from an elevation of 2600–3952 m in three days, it was reported that 59% met the criteria for AMS, which is greater than that observed in adults, and was correlated with altitude and upper respiratory infection within the seven days before their trek [12].

High-altitude cerebral edema (HACE) or high-altitude pulmonary edema (HAPE) may occur when progressive AMS is untreated or underdiagnosed. HACE is an emergent and life-threatening medical condition, in which the swelling of the brain happens due to HAI. The marked symptoms are ataxia, fatigue, and altered consciousness. HAPE is a fatal form of severe HAI. It often causes illness and is even fatal when diagnosed with HACE or HAPE [13]. Additionally, exposure to high altitude might bring about both physical and mental distress, inclusive of sleep problems, anxiety, and panic symptoms. Therefore, it is important to clarify the relationship between AMS and psychiatric morbidity.

Individuals who ascend to higher altitudes are prone to develop severe physical discomforts and AMS. Sleep disturbance might occur when the individuals are diagnosed with AMS or have altitude-related physical problems [14]. Some studies support the evidence that there is a reciprocal relationship between anxiety and AMS [3]. Psychotic disorders might also develop in high altitudes, which may well be with or without the HACE [15]. In addition, baseline somatization might also contribute to the HAI [16].

Although some researchers warned of a potential relationship between the psychological symptoms and AMS, there are only rare studies on the topic of AMS and the risk of subsequent psychiatric disorders. Therefore, we hypothesized that AMS may be related to the development of psychiatric disorders, and we aimed to explore if there is a relationship between AMS and the risk of psychiatric disorders in Taiwan by using the National Health Insurance Research Database (NHIRD).

## 2. Materials and Methods

### 2.1. Dataset Sources

The National Health Insurance (NHI) Program is a general and mandatory health insurance in Taiwan. The NHI Program includes contracts with 97% of the medical workers and began operating in 1995. It benefits more than 99% of the population; approximately 23 million people in Taiwan [17]. Some previous research recorded the details of this program [18]. There is comprehensive data regarding almost all the patients in the NHIRD. In our study, we included an inpatient database from 2000 to 2015 in the NHIRD. Participants were diagnosed with a code by the International Classification of Disease, Ninth Revision, Clinical Modification (ICD-9-CM).

### 2.2. Study Design and Patient Selection

This retrospective matched cohort study includes the inpatient data files from 1 January 2000 to 31 December 2015. Individual participants with AMS were defined as the patients diagnosed with the ICD codes as 993.2 and high altitude periodic breathing (ICD codes: 327.22), in an inpatient setting. The definitions of high-altitude cerebral edema (AMS plus cerebral edema, HACE) and high-altitude pulmonary edema (AMS plus pulmonary edema, HAPE) are defined as cerebral edema or pulmonary edema within one month after the index date of the diagnosis of AMS. A 1:10 matched for sex, age, insurance premium, comorbidities, location, level of care, and index date controls were randomly chosen for each patient with AMS. Individuals with unknown sex, without tracking, aged less than 20 years, and individuals diagnosed with psychiatric disorders or AMS before the index date were excluded in this study. The definition of the index date in this cohort was the time when the patient first received the diagnosis of AMS within the one-year study interval (Figure 1).

### 2.3. Outcomes

Both the AMS patients and the control group were followed from the index date until the end of 2015, or the development of psychiatric disorders, including dementia, anxiety disorders, depressive disorders, bipolar disorders, sleep disorders, posttraumatic stress disorder/acute stress disorder (PTSD/ASD), psychotic disorders, including schizophrenia and other psychotic disorders, and substance-related disorders (SRD), including alcohol usage disorder (AUD), and illicit drug usage disorder (IUD), or withdrawal from the NHI program. The ICD-9-CM codes of diagnoses in this study are as listed in Table 1. The diagnosis of psychiatric disorders must be confirmed by a board-certificated psychiatrist and meet the criteria from the Diagnostic and Statistical Manual of Mental Disorders-IV (DSM-IV), or the Diagnostic and Statistical Manual of Mental Disorders-IV-TR (DSM-IV-TR) in Taiwan.

### 2.4. Covariates

The covariates are sociodemographic characteristics and comorbidities. Sociodemographic characteristics included sex, age (20–49, 50–64, ≥65 years), monthly insured premiums, urbanization levels, regions of residence, and levels of medical care. There are three categories of monthly insured premiums in New Taiwan Dollars [NT$], including <18,000, 18,000–34,999 and ≥35,000. The urbanization level was determined by the population and the level of city development. Level 1 urbanization was determined as a population of more than 1,250,000 people. Level 2 urbanization was determined as a population between 500,000 and 1,250,000. Urbanization level 3 was determined as a population between 150,000 and 500,000. Urbanization level 4 was determined as a population less than 150,000. The Charlson comorbidity index (CCI) is an extensively used comorbidity index [19]. The CCI consists of 22 conditions [20], including myocardial infarction, congestive heart failure, peripheral vascular disease, dementia, cerebrovascular disease, chronic lung disease, connective tissue disease, ulcer, chronic liver disease, diabetes, hemiplegia, moderate or severe kidney disease, diabetes with end organ damage, tumor, leukemia, lymphoma, moderate or severe liver disease, malignant tumor, metastasis, and acquired immune deficiency syndrome.

### 2.5. Statistical Analysis

The statistical analyses were conducted by using the SPSS software version 22 (SPSS Inc., Chicago, IL, USA). Categorical variables were analyzed by using the Pearson chi-square test and continuous variables presented as the mean (±SD), were assessed utilizing the two-sample t test. The difference in the cumulative survival of the psychiatric disorders for the patients with AMS and the controls was calculated by using the Kaplan–Meier method with the log-rank test. We analyzed the competing risk and measured the sub-distribution hazard ratios (sHRs) and 95% confidence interval (CI), adjusting the covariates by utilizing the Fine and Gray’s model (competing with mortality) in order to explore the association between the development of psychiatric disorders and the experience of AMS. The Fine and Gray’s survival analysis was conducted by utilizing the value-added module, including the competing risks survival analysis, in the SPSS. (https://www.asiaanalytics.com.tw/en/product/p-asia-analytics-2.jsp, accessed on 21 October 2021).

The study used the Bonferroni method for multiple comparisons, and set the *p* value below 0.001 as being statistically significant.

## 3. Results

### 3.1. Study Cohort Characteristics

Table 2 reveals the sex, age, monthly insured premiums, comorbidities, seasons for seeking medical help, residences, urbanization level, and levels of medical care in the AMS cohort and the control group. There is no significant difference in the covariates when compared to the patients with AMS and the control group.

### 3.2. Kaplan–Meier Curves for the Cumulative Survival of Psychiatric Disorders

In total, there were 127 patients diagnosed with AMS in the study interval. There were 49 patients in the AMS cohort (N = 127) and 140 patients in the controls (N = 1270) who were diagnosed with psychiatric disorders (4175.58 vs. 1126.19 per 100,000 person-years) during the follow-up period. Figure 2 showed that there was a significant difference of cumulative survival between the two cohorts in the psychiatric disorders (log-rank test, *p* < 0.001).

### 3.3. Sub-Distribution Hazard Ratio Analysis of Psychiatric Disorders in the AMS Group

Table 3 reveals the factors associated with the risk of psychiatric disorders after AMS by using Fine and Gray’s survival analysis. The crude sHR was 9.999 (95% CI: 7.123–14.036, *p* < 0.001) for psychiatric disorders. The adjusted sHR was 10.384 (95% CI: 7.267–14.838, *p* < 0.001) after adjusting for sex, age, monthly insured premiums, urbanization levels, residences, comorbidities, seasons for seeking medical help, and levels of medical care. Those in the AMS group that obtained medical care from the medical center were prone to have a greater risk for the development of psychiatric disorders.

### 3.4. Subgroup Analysis of Psychiatric Disorders in the AMS Group and the Control Group

Table 4 reveals that the AMS group was prone to have a higher risk for the development of psychiatric disorders than the controls, despite the sex, age, monthly insured premiums, comorbidities, seasons for seeking medical help, residences, urbanization level, and levels of care.

### 3.5. Sensitivity Analysis and Types of Psychiatric Disorders

Appendix A indicates that AMS is associated with the overall psychiatric disorders, anxiety, depressive, bipolar, sleep disorder, PTSD/ASD, psychotic disorders, including schizophrenia, other psychotic disorders, and SRD, including AUD and IUD. After we excluded the diagnoses within the first year after the index date, AMS was associated with anxiety, depression, bipolar, sleep disorder, PTSD/ASD, psychotic disorders, including schizophrenia and other psychotic disorders, and SRD, including AUD and IUD. Nonetheless, after we excluded the diagnoses within the first five years after the index date, AMS was associated with the overall psychiatric disorders, anxiety, depression, sleep disorder, and SRD, including AUD.

The AMS cohort had a greater risk of the development of PTSD/ASD: the risk of progression to PSTD was within the first five years. In addition, the AMS cohort had a greater risk of the development of SRD. The AMS cohort had around a 14-fold rising risk of developing to SRD and around an 8-fold rising risk of the development of AUD after we excluded the psychiatric disorders within the first five years. The AMS cohort had around a 29.7-fold rising risk of developing to bipolar disorder and around a 7.6-fold rising risk of the development of bipolar disorder after we excluded the psychiatric disorders within the first year. The AMS cohort had around a 21-fold rising risk of developing depression, around a 15-fold rising risk of the development of depression after we excluded the psychiatric disorders within the first year, and around a 12-fold rising risk of the development of depression after we excluded the psychiatric disorders within the first five years. The AMS cohort had around a 17.5-fold rising risk of developing anxiety, around a 12-fold rising risk of the development of anxiety after we excluded the psychiatric disorders within the first year, and around a 23-fold rising risk of the development of anxiety after we excluded the psychiatric disorders within the first five years. The AMS cohort had around an 8.5-fold rising risk of developing sleep disorders, around an 8.5-fold rising risk of the development of sleep disorders after we excluded the psychiatric disorders within the first year, and around a 6-fold rising risk of the development of sleep disorders after we excluded the psychiatric disorders within the first five years. Thus, the result should serve as a reminder to doctors to monitor the psychological health of the AMS patients.

### 3.6. Years from AMS to the Development of Psychiatric Disorders

The mean interval from the index date to the development of psychiatric disorders was 5.17 (SD [standard deviation] = 4.26) years. The mean years to the development of psychiatric disorders in patients with AMS were 2.96 (SD = 3.56) years, which was shorter than the control group (5.95 [SD = 4.23] years) (Appendix A).

### 3.7. AMS and the Risk of Single or Multiple Psychiatric Diagnoses

For the potential of comorbid in the psychiatric disorders, we analyzed the association between AMS and single or multiple psychiatric diagnoses. Appendix A revealed that AMS is associated with either single (adjusted hazard ratio [HR] = 9.890, 95%: 6.921–14.131, *p* < 0.001) or multiple psychiatric disorders (adjusted HR = 10.789, 95%: 7.550–15.416, *p* < 0.001).

## 4. Discussion

There are numerous meaningful findings in this study: First, there is a nearly 10-fold risk of the development to overall psychiatric disorders when comparing the AMS group to the controls. To the best of our knowledge, this is the first study from a nationwide, population-based database, in a 16-year follow-up, on the relationship between AMS and the rising risk of progressing to psychiatric disorders.

Second, AMS is related to the wide range of psychiatric disorders and the relationship between anxiety, depression, sleep disorders, and SRD, and AMS still persisted even after we excluded the psychiatric disorders within the first five years after AMS. In addition, AMS is associated with the risk of both single and multiple psychiatric diagnoses.

Third, the mean years to the development of psychiatric disorders in AMS patients were 2.96 (±3.56) years.

In addition, the AMS cohort had a greater risk of the development of SRD. These findings suggest that there is a long-term influence of AMS on the psychological health of the patients. Furthermore, according to the sensitivity analysis, the protopathic bias from an undiagnosed disorder could be avoided in this study [21].

Fourth, the present study depicts the association between AMS and the subsequent development of a broad range of psychiatric disorders, instead of the relationship between AMS and one or two neuropsychiatric disorders, such as irreversible subcortical dementia developed following AMS [22], and psychotic disorders [15]. However, in this study, we did not find the association between AMS and the development of dementia. The reasons remain unclear and therefore need to be studied. In addition, previous studies found that some personality traits [23] and somatization [16] might also contribute to the HAI.

Hüfner et al., (2022) suggested that individuals with previous psychiatric history revealed higher functional AMS scores. However, there were some individuals who experienced AMS and had no previous diagnosis of psychiatric disorder, but reported a history compatible with psychiatric disorder when questioned directly by the physician, which may be associated with stigmatization or sociocultural influences or recall bias. Higher AMS scores with symptoms of nausea, dizziness, and weakness may overlap with symptoms of depression and anxiety [5]. In addition, anxious individuals are more likely to develop AMS [3,24]. It is possible that pre-existing symptoms act as confounder for the Lake Louise acute mountain sickness score [25]. Thus, the false positive AMS scoring owing to the imminent psychiatric symptoms should be considered. It is possible that the common underlying pathophysiology of psychiatric disorder may be a “false positive” AMS scoring due to the imminent psychiatric symptoms.

The fundamental mechanisms of the development to psychiatric disorders after AMS are still unclear. Fatigue, breathlessness, and insomnia are common symptoms related to the increasing high altitudes, and exposure potentially triggers greater stress and potential anxiety. In point of fact, evidence suggested that increasing anxiety may result in greater somatization and being prone to developing AMS at high altitude [3]. One study revealed that sleep disturbance may indicate a potential sign for the development of bipolar and unipolar episodes [26]. Low-grade inflammatory responses [27] and the cortisol level increase [28] might be a possible pathophysiology between high altitude and mood changes. Furthermore, the serotonin level is decreased at high altitude in animal subjects [29].

There were some possible mechanisms illustrating high altitude-related psychosis. Several studies found that hypoxia, hypoglycemia, and cold induced dysfunction of the angular gyrus and temporoparietal junction may be associated with high altitude-related psychosis [30]. Furthermore, social and sensory deprivation and psychosocial pressure may result in high altitude-related psychosis [15]. The white matter changes that resulted from HACE in patients having the risk of schizophrenia, especially in the splenium of the corpus callosum may be related to the development of psychosis [31]. PTSD can develop after mountain-related traumatic experiences to oneself or team members during mountain climbing with subsequent injury [32].

There is a prominent finding on the occurrence of depression and anxiety after experiencing AMS. The AMS cohort had around a 23-fold rising risk of the development of anxiety after we excluded the psychiatric disorders within the first five years. The AMS cohort had around a 12-fold rising risk of the development of depression after we excluded the psychiatric disorders within the first five years. Excessive fear and anxiety may occur when immediate medical care is not available. Individuals had more negative moods after experiencing AMS [33]. New onset of anxiety related disorders, such as panic attacks, excessive health-related anxiety, and excessive emotionality may occur after being affected by AMS, which may be associated with hyperventilation, periodic breathing, sleep disturbance, and environmental adjustment [34].

In this study, patients with AMS care from the medical centers were prone to have a rising risk for the development of psychiatric disorders. It is possible that patients with severe AMS tended to be sent to a hospital center, and the increased severity of AMS may be related to the increased risk of the occurrence of psychiatric disorders. However, further research is necessary to explore the fundamental mechanisms for the occurrence of psychiatric disorders after experiencing AMS.

### 4.1. Strengths of This Study

There are several strengths in this study: First, we utilized the Longitudinal Health Insurance Database (LHID), which has a prodigious number of samples in this study. Second, there were several studies that considered the relationship between the preexisting psychiatric problems and the incidence of AMS [3,4,14,16]. However, a limited study discussed the association between AMS and the following of psychiatric risks, and the mechanism is still unclear. Third, this is a long-term follow-up study of 16 years. We aim to explore if there is a relationship between the development of psychiatric disorders and the experiment of AMS. The psychological impact in AMS might not only be from the initial state of acute anxiety [35], psychosis [15], and acute trauma-related psychiatric disorders, such as ASD or acute PTSD [36]. The monitoring of psychiatric disorders for patients after AMS for a longer period of time would be of significant importance.

### 4.2. Limitations of This Study

This study has some limitations. First, in the NHIRD, there was no information on the severity and laboratory data in the AMS patients, which is similar to previous research using the NHIRD [37,38,39,40,41]. Second, genetic, environmental, and psychosocial factors were not presented in the database. Third, it is likely to have ascertainment bias if the patients treated for AMS were also consulted for psychiatric situations. Furthermore, there are very few cases in this study. Only a few numbers of bipolar disorder, PTSD, and psychotic disorder occurred within five years after AMS. Furthermore, only one developed psychiatric disease after 12 years in the AMS group, which indicated there was little difference between the two groups after 12 years in the research period (Appendix A). Thus, we should pay more attention to the development of psychiatric disorders within 12 years after AMS. After all, the diagnoses and the subsequent treatment of the development of psychiatric consequences after AMS is limited due to the small number of AMS cases. Furthermore, AMS would be usually self-resolved by this time. In addition, some individuals with HACE, or beginning HACE, might not be brought to medical attention. Therefore, some AMS patients would not enter into the database of health insurance, since some AMS individuals need not be rescued. This issue might influence the analysed data. Moreover, there is no information from the AMS patients’ family members and the health staff who looked after them in the NHIRD. Further studies to explore the post-AMS psychiatric comorbidity are needed in the future follow-up.

## 5. Conclusions

This study found that there was an association between AMS and the rising risk of psychiatric disorders in the 16 years of a long-term follow-up study. This should serve as a reminder to the physicians that it is an essential concern on the psychiatric comorbidity of the patients with AMS.

## Figures and Tables

**Figure 1 ijerph-20-02868-f001:**
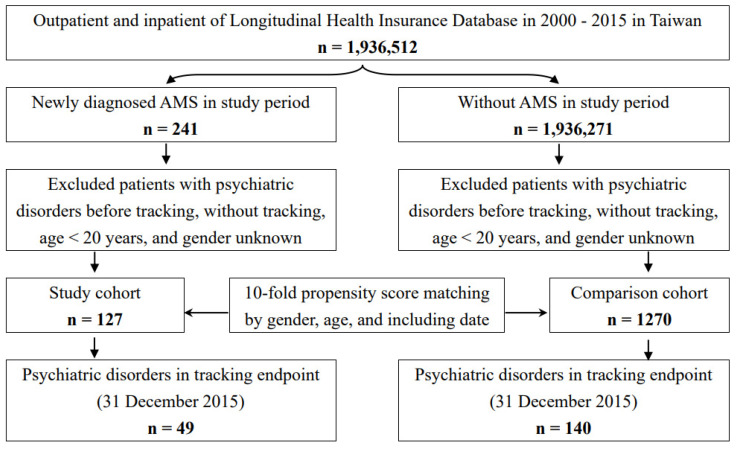
The flowchart of study sample selection. The arrows mean the selection process in the sample. Abbreviations: AMS, acute mountain sickness.

**Figure 2 ijerph-20-02868-f002:**
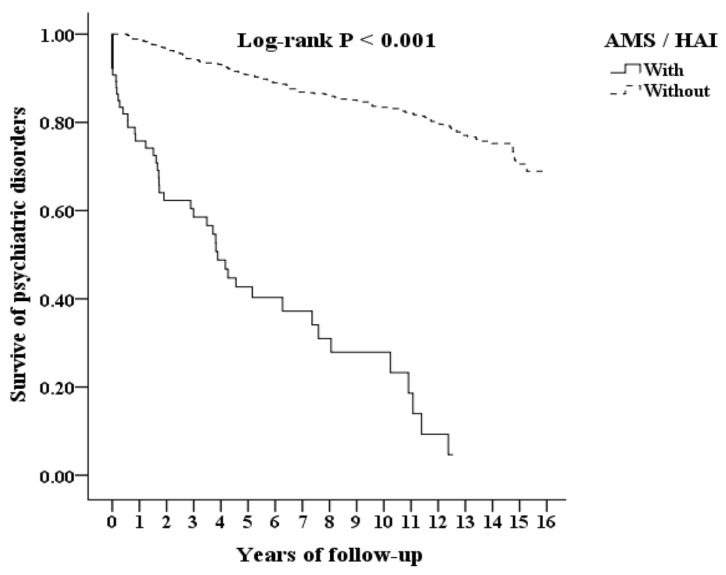
Kaplan–Meier for cumulative survival of psychiatric disorders among those aged 20 and over-stratified by acute mountain sickness (AMS) with a log-rank test among the AMS cohort and the control group. The solid line means the cumulative survival of psychiatric disorders of the patients with AMS. The dotted line means the cumulative survival of psychiatric disorders of the patients without AMS. There was a significant difference in cumulative survival between the two cohorts in the psychiatric disorders (log-rank test, *p* < 0.001). Abbreviations: AMS, acute mountain sickness.

**Table 1 ijerph-20-02868-t001:** ICD-9-CM codes of diagnoses in this study.

Acute Mountain Sickness	993.2
High altitude periodic breathing	327.22
Acute mountain sickness only	993.2 only
High altitude cerebral edema	993.2 + 348.5 (within 1 month after index date)
High altitude pulmonary edema	993.2 + 518.4 (within 1 month after index date)
Psychiatric disorders	
Dementia	290.0, 290.10–290.13, 290.20–290.21, 290.3, 290.41–290.43, 290.8–290.9, 331.0
Anxiety disorders	300
Depressive disorders	296.2–296.3, 300.4, 311
Bipolar disorders	296.0, 296.4–296.8
Sleep disorders	307.4, 780.5
Posttraumatic stress disorder/acute stress disorder	308, 309.81
Psychotic disorders	295, 297–298
Schizophrenia	295 except 295.4
Schizophreniform disorder	295.4
Other psychotic disorders	297–298
Substance-related disorders	291–292, 303.0, 303.9, 304–305
Alcohol use disorder	291, 303.0, 303.9, 305.0
Illicit drug use disorder	292, 304–305 except 305.0

Abbreviations: ICD-9-CM, International Classification of Diseases, Ninth Revision, Clinical Modification.

**Table 2 ijerph-20-02868-t002:** Characteristics of study at the baseline among the acute mountain sickness cohort and the control group.

AMS	With (*n* = 127)		Without (*n* = 1270)		*p*
Variables	*n*	%	*n*	%	
Total	127	9.09	1270	90.91	
Gender					0.999
Male	90	70.87	900	70.87	
Female	37	29.13	370	29.13	
Age (years)	51.85 ± 17.82		51.30 ± 18.11		0.748
Age groups (yrs)					0.999
20–49	48	37.80	480	37.80	
50–64	45	35.43	450	35.43	
≥65	34	26.77	340	26.77	
Insured premium (NT$)					0.111
<18,000	123	96.85	1240	97.64	
18,000–34,999	2	1.57	26	2.05	
≥35,000	2	1.57	4	0.31	
CCI_R	0.41 ± 1.07		0.70 ± 1.67		0.055
CCI_R groups					0.116
0	99	77.95	887	69.84	
1	15	11.81	192	15.12	
2	9	7.09	72	5.67	
3	2	1.57	64	5.04	
≥4	2	1.57	55	4.33	
Season					0.999
Spring (Mar–May)	19	14.96	190	14.96	
Summer (Jun–Aug)	42	33.07	420	33.07	
Autumn (Sep–Nov)	43	33.86	430	33.86	
Winter (Dec–Feb)	23	18.11	230	18.11	
Location					0.283
Northern Taiwan	59	46.46	477	37.56	
Middle Taiwan	34	26.77	385	30.31	
Southern Taiwan	26	20.47	334	26.30	
Eastern Taiwan	8	6.30	69	5.43	
Outlets islands	0	0.00	5	0.39	
Urbanization level					0.094
1 (The highest)	37	29.13	436	34.33	
2	64	50.39	515	40.55	
3	6	4.72	119	9.37	
4 (The lowest)	20	15.75	200	15.75	
Level of care					<0.001
Hospital center	56	44.09	364	28.66	
Regional hospital	47	37.01	392	30.87	
Local hospital	24	18.90	514	40.47	

There is no significant difference in the covariates when comparing the patients with AMS to the control group. Abbreviations: AMS, acute mountain sickness; CCI, Charlson comorbidity index; NT$, new Taiwan dollars. *P* Chi-square/Fisher exact test on category variables and *t*-test on continue variables.

**Table 3 ijerph-20-02868-t003:** Factors of psychiatric disorders by using Fine and Gray’s competing risk model among the acute mountain sickness cohort and the control group.

	Competing Risk in the Model							
Variables	Crude sHR	95% CI	95% CI	*p*	Adjusted sHR	95% CI	95% CI	*p*
AMS (reference: without AMS)	9.999	7.123	14.036	<0.001	10.384	7.267	14.838	<0.001
Male (reference: female)	1.212	0.883	1.665	0.234	1.386	0.998	1.924	0.051
Age 50–64 (reference: age 20–49)	1.846	0.584	2.226	0.378	1.791	0.520	2.202	0.273
Age ≥ 65 (reference: age 20–49)	1.915	0.616	2.359	0.659	1.805	0.541	2.297	0.283
Insured premium (NT$) 18,000–34,999 (reference: Insured premium < 18,000)	3.218	0.798	12.976	0.100	3.262	0.750	14.192	0.115
Insured premium (NT$) ≥ 35,000 (reference: Insured premium < 18,000)	0.378	0.094	1.522	0.171	0.403	0.099	1.634	0.203
CCI = 1 (reference: CCI = 0)	1.020	0.715	1.456	0.912	0.953	0.659	1.379	0.799
CCI = 2 (reference: CCI = 0)	1.118	0.683	1.830	0.657	1.328	0.790	2.233	0.284
CCI = 3 (reference: CCI = 0)	0.991	0.565	1.737	0.974	1.084	0.604	1.944	0.787
CCI ≥ 4 (reference: CCI = 0)	0.814	0.470	1.410	0.463	0.994	0.650	2.050	0.625
Season Summer (reference: spring)	1.512	1.004	2.279	0.048	1.509	0.993	2.294	0.054
Season Autumn (reference: spring)	1.194	0.767	1.858	0.433	1.310	0.835	2.054	0.240
Season Winter (reference: spring)	1.245	0.797	1.946	0.335	1.163	0.736	1.835	0.518
Urbanization level								
1 (The highest)	1.150	0.768	1.721	0.498	1.091	0.693	1.719	0.706
2	1.146	0.663	1.980	0.625	1.049	0.585	1.879	0.873
3	0.802	0.565	1.140	0.220	0.809	0.580	0.170	0.260
4 (The lowest)	Reference				Reference			
Medical center (reference: Local hospital)	1.800	1.228	2.639	0.003	1.698	1.092	2.639	0.019
Regional hospital (reference: Local hospital)	1.387	0.974	1.974	0.070	1.168	0.785	1.739	0.443

Abbreviations: AMS, acute mountain sickness; CCI, Charlson comorbidity index; NT$, new Taiwan dollars; sHR, sub-distribution hazard ratio; *P* Chi-square/Fisher exact test on category variables and *t*-test on continue variables; CI, confidence interval; and competing variable: all-cause mortality.

**Table 4 ijerph-20-02868-t004:** Subgroup analysis of the factors for the risk of psychiatric disorder development in acute mountain sickness.

AMS	No competing Risk in the Model				Competing Risk in the Model			
Stratified	Adjusted HR	95% CI	95% CI	*p*	Adjusted sHR	95% CI	95% CI	*p*
Total	10.609	7.424	15.159	<0.001	10.384	7.267	14.838	<0.001
Gender								
Male	10.855	7.596	15.510	<0.001	10.625	7.435	15.182	<0.001
Female	10.182	7.125	14.549	<0.001	9.966	6.975	14.241	<0.001
Age groups (years)								
20–49	10.057	7.038	14.370	<0.001	9.844	6.889	14.066	<0.001
50–64	10.283	7.196	14.692	<0.001	10.064	7.043	14.381	<0.001
≥65	11.180	7.823	15.974	<0.001	10.942	7.658	15.636	<0.001
Insured premium (NT$)								
<18,000	10.296	7.205	14.711	<0.001	10.077	7.052	14.400	<0.001
18,000–34,999	55.592	38.902	79.434	<0.001	54.413	38.079	77.752	<0.001
≥35,000	6.551	4.584	9.360	<0.001	6.412	4.487	9.162	<0.001
CCI_R groups								
0	12.043	8.428	17.208	<0.001	11.788	8.249	16.844	<0.001
1	6.729	4.709	9.615	<0.001	6.587	4.609	9.412	<0.001
2	18.686	13.076	26.701	<0.001	18.290	12.800	26.135	<0.001
3	19.960	13.968	28.521	<0.001	19.537	13.673	27.917	<0.001
≥4	7.257	5.079	10.370	<0.001	7.104	4.971	10.150	<0.001
Season								
Spring	6.886	4.818	9.839	<0.001	6.740	4.717	9.630	<0.001
Summer	15.289	10.699	21.846	<0.001	14.965	10.473	21.383	<0.001
Autumn	12.662	8.861	18.092	<0.001	12.393	8.673	17.709	<0.001
Winter	7.265	5.084	10.381	<0.001	7.111	4.976	10.161	<0.001
Urbanization level								
1 (The highest)	13.753	9.624	19.652	<0.001	13.462	9.421	19.236	<0.001
2	13.398	9.376	19.144	<0.001	13.114	9.178	18.739	<0.001
3	5.195	3.636	7.424	<0.001	5.085	3.559	7.266	<0.001
4 (The lowest)	5.624	3.936	8.036	<0.001	5.505	3.852	7.866	<0.001
Level of care								
Medical center	28.888	20.215	41.277	<0.001	28.275	19.788	40.403	<0.001
Regional hospital	10.378	7.262	14.828	<0.001	10.158	7.109	14.514	<0.001
Local hospital	5.124	3.586	7.322	<0.001	5.016	3.510	7.167	<0.001

Abbreviations: HR, Hazard ratio; sHR, sub-distribution hazard ratio: adjusted for the variables listed in Table 2; CCI, Charlson comorbidity index; NT$, new Taiwan dollars; CI, confidence interval; and competing variable: all-cause mortality.

## Data Availability

All data are presented within the article.

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
