# Peer review of "Acute Mountain Sickness and the Risk of Subsequent Psychiatric Disorders—A Nationwide Cohort Study in Taiwan"

_ijerph, 2023, doi:10.3390/ijerph20042868_

Round 1

Reviewer 1 Report

This is a very interesting and novel manuscript reporting on a very recent observation i.e. the association of mental disorders with AMS. This was only recently described in a first manuscript. (Please see Hüfner K, Caramazza F, Pircher Nöckler ER, Stawinoga AE, Fusar-Poli P, Bhandari SS, Basnyat B, Brodmann Maeder M, Strapazzon G, Tomazin I, Zafren K, Brugger H, Sperner-Unterweger B. Association of Pre-existing Mental Health Conditions with Acute Mountain Sickness at Everest Base Camp. High Alt Med Biol. 2022 Sep 7. doi: 10.1089/ham.2022.0014. Epub ahead of print. PMID: 36070557.). This first report on the concept investigated in the present manuscript should be referenced. The idea of the study is great and of relevance.

Main point of concern:

AMS is usually not a diagnosis entered into the database of a health insurance because unless individuals need to be rescued, which is very rare in AMS, the condition will be self-resolved by the time individuals reach any medical care points.

Mostly individuals with HACE or beginning HACE are probably brought to medical attention outside of field studies. This might greatly influence the analysed data.

Further points:

Figure 1 is confusing with the arrows crossing each other

Considering the number of comparisons a correction for multiple comparisons should be done. Also for t test data distribution should be assessed.

The discussion has a very long section (multiple paragraphs) summarizing the main findings. These should be left in the results section and only a short summary (5-6 sentences) at the beginning of the Discussion should summarize the main findings.

It does not really make sense to discuss in detail the occurrence of PTSD if only so few cases (if I understand correctly only 2 in one and 1 in the other group) occurred. I think these calculations are only valid for the groups with more cases in them, so the authors should concentrate on interpreting the more robust findings in more detail.

In my opinion too little is written on the possible underlying pathomechanisms. I think the main point here is a common underlying pathophysiology vs a “false positive” AMS scoring due to imminent psychiatric symptoms. This should in my opinion be the main topic of the discussion, rather than going into details of specific disorders which were detected at very low frequencies.

Reviewer 2 Report

The article presents very interesting and clinically significant data on association between acute mountain sickness and risk of development of psychiatric disorders in a 16-year follow up.

Overall, the article is well-written. I only recognise minor issues to be addressed prior to publication

In Abstract, the AMS abbreviation is not explained upon its first use - please correct it.

Figurę 2 - I suggest moving the table below the plot into the Supplementary Material

Table 5, 6,7  - I suggest moving it into the Supplementary Material

Please make sure all the abbreviations are explained upon their first use and in the Tables and Figures

The descriptions of the Tables and Figures should be more precise, i.e. they should include a short characteristics of the group, .e.g. „among the acute mountain sickness cohort and the control group

A generał workup by a language editor is recommended, preferably a native speaker.
